# Osteonecrosis of the Femoral Head Safely Healed with Autologous, Expanded, Bone Marrow-Derived Mesenchymal Stromal Cells in a Multicentric Trial with Minimum 5 Years Follow-Up

**DOI:** 10.3390/jcm10030508

**Published:** 2021-02-01

**Authors:** Enrique Gómez-Barrena, Norma G. Padilla-Eguiluz, Philippe Rosset, Philippe Hernigou, Nicola Baldini, Gabriela Ciapetti, Rosa M. Gonzalo-Daganzo, Cristina Avendaño-Solá, Hélène Rouard, Rosaria Giordano, Massimo Dominici, Hubert Schrezenmeier, Pierre Layrolle

**Affiliations:** 1Servicio de Cirugía Ortopédica y Traumatología, Hospital Universitario La Paz-IdiPAZ, 28046 Madrid, Spain; 2Facultad de Medicina, Universidad Autónoma de Madrid, 28029 Madrid, Spain; 3Universidad Autónoma de Madrid-IdiPAZ, 28049 Madrid, Spain; norma.padilla@uam.es; 4Service de Chirurgie Orthopédique et Traumatologique 2, Hôpital Trousseau, Université François-Rabelais de Tours, CHU de Tours, 37044 Tours, France; philippe.rosset@univ-tours.fr; 5Orthopaedic Department, Hôpital Henri Mondor, InsermU955, 94000 Créteil, France; 6Department of Orthopaedic Surgery, Faculty of Medicine, UPEC (University Paris-Est, Créteil), 94000 Créteil, France; philippe.hernigou@wanadoo.fr; 7Department of Biomedical and Neuromotor Sciences, University of Bologna, 40136 Bologna, Italy; nicola.baldini@ior.it; 8SC BST, IRCCS Istituto Ortopedico Rizzoli, 40136 Bologna, Italy; gabriela.ciapetti@ior.it; 9Servicio de Hematología, Hospital Universitario Puerta de Hierro-Majadahonda, 28222 Madrid, Spain; rosamaría.gonzalo@salud.madrid.org; 10Servicio de Farmacología Clínica, Hospital Universitario Puerta de Hierro-Majadahonda, and Universidad Autónoma de Madrid, 28222 Madrid, Spain; cavendano@salud.madrid.org; 11Établissement Français du Sang, 94000 Paris, France; helene.rouard@efs.sante.fr; 12Laboratory of Regenerative Medicine—Cell Factory, Transfusion Center, Fondazione IRCCS Ca’ Granda, Ospedale Maggiore Policlinico, 20122 Milano, Italy; rosaria.giordano@policlinico.mi.it; 13Laboratory of Cellular Therapies, Department of Medical and Surgical Sciences for Children & Adults, University—Hospital of Modena and Reggio Emilia, 41121 Modena, Italy; massimo.dominici@unimore.it; 14Institut for Transfusion Medicine, Ulm University, and Institute for Clinical Transfusion Medicine and Immunogenetics Ulm, German Red Cross Blood Transfusion Service and University Hospital Ulm, 89081 Ulm, Germany; h.schrezenmeier@blutspende.de; 15INSERM U957, Lab. Pathophysiology of Bone Resorption, Faculty of Medicine, University of Nantes, 44035 Nantes, France; pierre.layrolle@inserm.fr

**Keywords:** femoral head osteonecrosis, bone regeneration, expanded MSC, clinical trial

## Abstract

*Background*: Osteonecrosis (ON) of the femoral head represents a potentially severe disease of the hip where the lack of bone regeneration may lead to femoral head collapse and secondary osteoarthritis, with serious pain and disability. The aim of this European, multicentric clinical trial was to prove safety and early efficacy to heal early femoral head ON in patients through minimally invasive surgical implantation of autologous mesenchymal stromal cells (MSC) expanded from bone marrow (BM) under good manufacturing practices (GMP). *Methods*: Twenty-two patients with femoral head ON (up to ARCO 2C) were recruited and surgically treated in France, Germany, Italy and Spain with BM-derived, expanded autologous MSC (total dose 140 million MSC in 7 mL). The investigational advanced therapy medicinal product (ATMP) was expanded from BM under the same protocol in all four countries and approved by each National Competent Authority. Patients were followed during two years for safety, based on adverse events, and for efficacy, based on clinical assessment (pain and hip score) and imaging (X-rays and MRIs). Patients were also reviewed after 5 to 6 years at latest follow-up for final outcome. *Results*: No severe adverse event was recalled as related to the ATMP. At 12 months, 16/20 per protocol and 16/22 under intention-to-treat (2 drop-out at 3 and 5 months) maintained head sphericity and showed bone regeneration. Of the 4 hips with ON progression, 3 required total hip replacement (THR). At 5 years, one patient (healed at 2 years visit) was not located, and 16/21 showed no progression or THR, 4/21 had received THR (all in the first year) and 1 had progressed one stage without THR. *Conclusions*: Expanded MSCs implantation was safe. Early efficacy was confirmed in 80% of cases under protocol at 2 years. At 5 years, the overall results were maintained and 19% converted to THR, all in the first year.

## 1. Introduction

Osteonecrosis (ON) of the femoral head, whether idiopathic or secondary to alcoholism, steroids, sickle cells or other conditions, is a devastating health problem with no current cure. The established necrosis of the femoral head bone may progress to head collapse with serious pain and functional limitation. Its pathophysiology includes femoral head destruction followed by reparative processes in the periphery of the necrotic area [1]. Despite some spontaneous bone regeneration in a mineralization front that isolates the necrotic area, bone resorption initiates the subchondral fracture that may lead to femoral head collapse [2]. In the early phases of this condition, while the femoral head sphericity is maintained, different treatments have been proposed, with variable effectiveness [3] to enhance bone regeneration, trying to avoid head collapse [4]. Total hip replacement (THR) is currently increasing not only as a final treatment but at any stage [5], even in the young patient, with obvious disadvantages specially at early ages.

Treatment of early osteonecrosis stages is initially based in a drill or forage into the necrotic area of the femoral head, with limited effectiveness except in early ON stages [6,7]. Although originally this forage was considered a means of releasing high pressure within the bone compartment (core decompression), it has evolved into a transosseus pathway to reach the necrotic area of the femoral head and deliver biotechnology and regenerative agents [8].

The development of alternative solutions to enhance healing through bone regeneration is a challenging aim. Although, overall, regenerative medicine techniques (including mesenchymal stem cell (MSC) implantation in the osteonecrotic area, and others such as intra-arterial infiltration with MSC, implantation of bioactive molecules, or even platelet-rich plasma) are seen as a means to avoid THR [9], the most suitable patients and the most appropriate procedures are still unclear.

Medicaments based on culture-expanded autologous mesenchymal stromal cells (MSC) may fulfil the aim of delivering a high number of cells, capable of generating new bone [10], with a potential role in osteoinduction and osteogenesis within the osteonecrotic femoral head. Early clinical data supporting expanded MSC implantation [11] unfortunately lack validated Good Manufacturing Practices (GMP) procedures [12] and appropriate quality controls to confirm safety and reproducibility of the cell medicament production and transportation [13]. Unless these data are offered to grant the regulatory approval, clinical application may not be accepted.

In this context, a European consortium designed this multicentric clinical trial with the primary aim of assessing the safety of expanded autologous bone-marrow-derived MSC (BM-MSC) implanted through forage into the femoral head. A secondary aim was set in the evaluation of efficacy to obtain bone healing in patients with early osteonecrosis of the femoral head at 1 and 2 years. To gain clinical relevance, we spread the aim of this clinical trial to also report in this paper the efficacy evaluation after 5 years.

## 2. Methods

### 2.1. Study Design and Participants

A phase I/IIa open, prospective, multicentric, interventional clinical trial named ORTHO2 in the REBORNE EU-funded project (Regenerating Bone defects using New biomedical Engineering approaches, FP7 HEALTH-2009-1.4-2, Grant Agreement 241879) was designed to evaluate safety and feasibility in five European centres from four countries, with total recruitment of 26 patients, 22 of whom received treatment (Figure 1, CONSORT diagram) from March 2014 to June 2015.

The inclusion criteria were age 18 to 65, both sexes, with symptomatic osteonecrosis of the femoral head, with less than 6 months of evolution, in patients that signed informed consent being able to provide it and able to understand and accept the study constraints, having medical health care coverage in any of the participating countries.

The exclusion criteria were pregnancy, breastfeeding women and women who are of childbearing age and not practising adequate birth control; participation in another therapeutic trial in the previous 3 months; stages 3 (III) or more (Arlet and Ficat, ARCO, Steinberg) of severe femoral head osteonecrosis, primarily based on diagnosis by imaging (X-rays, MRI); flattening or collapse of the femoral head (Arlet and Ficat 4, ARCO IV, Steinberg stage IV and beyond) or articular cartilage collapse at the time of core decompression surgery; septic arthritis; stress fracture; non-osteonecrosis metabolic bone diseases (particularly Paget’s disease of bone, osteogenesis imperfecta, primary hyperparathyroidism, fibrous dysplasia monostotic, polyostotic (McCune–Albright syndrome) and osteopetrosis); any active bisphosphonate treatment or any history of intravenous bisphosphonate treatment; history of prior or concurrent diagnosis of HIV-, hepatitis-B- or hepatitis-C-infection (confirmed by serology or PCR); active hepatitis B or hepatitis C infection at the time of screening; known allergies to products involved in the production process of MSC; acute and persistent chronic bacterial infections such as brucellosis, typhus, leprosy, relapsing fever, melioidosis and tularaemia; history of neoplasia or current neoplasia in any organ; corticoid or immunosuppressive therapy more than one week in the two months before study inclusion; patients who will require continuous, systemic, high-dose corticosteroid therapy (more than 7.5 mg/day) within 6 months after surgery; patients who are in active treatment for cancer or blood dyscrasia, or have received chemotherapy, radiotherapy or immunotherapy in the past 2 years; history of regular alcohol consumption exceeding 2 drinks/day (1 drink = 150 mL of wine or 360 mL of beer or 45 mL of hard liquor) within 6 months of screening and/or history of illicit drug use; serum AST (SGOT)/ALT (SGPT) > 2.5 × (institutional standard range); MRI-incompatible internal devices (pacemakers, aneurysm clips, etc.); body mass index (BMI) of 40 kg/m^2^ or greater; patients unable to tolerate general anaesthesia; insulin-dependent diabetes; patients with poorly controlled diabetes mellitus (HbA1C > 8%), or with peripheral neuropathy, or known concomitant vascular problems; patients receiving treatment with hematopoietic growth factors or anti-vasculogenesis or anti-angiogenesis treatment; traumatic osteonecrosis; adult in the care of a guardian (subject legally protected); or impossibility to meet at the appointments for the clinical follow-up.

The anonymous clinical data and imaging of the eligible patient were forwarded to the other clinical centres. Each patient was only included if no centre was against inclusion and at least two more centres agreed on the inclusion and the treatment.

To further standardize the inclusion, patients to be included were those who, alternatively, would have received forage, thus avoiding (excluding) patients with head collapse or sphericity loss to be treated with other reconstruction techniques.

As shown in the CONSORT diagram (Figure 1), 26 patients were recruited, one of them excluded before bone marrow extraction as hepatitis B virus-positive serology was confirmed. Three more were excluded before surgery, two due to the fact that the number of expanded cells did not achieve the required concentration, and one due to contamination of the investigational product.

Of the 22 treated patients and analysed under ITT (intention-to-treat), two of them caused early drop-out, the first one was imprisoned (at three months after surgery), and the second one was treated in another centre (at five months by the patient’s decision to undergo Total Hip Replacement).

Among the 22 treated patients, 1 case received treatment with a protocol deviation due to overgrowth of cells that required early release and implantation of the Investigational Medical Product (IMP) before Standard Operating Procedure (SOP) specification (15 days post bone marrow extraction, instead of 21 days).

The efficacy follow-up was performed at 12 months, and the safety follow-up at 24 months. Intermediate results were evaluated at three and six months. This communication also shows a late follow-up of 5 to 6 years.

A total of 22 patients were evaluated at three months, a total of 21 were analysed at six months and 12 months on ITT, and a total of 17 patients underwent follow-up at 5 to 6 years.

Table 1 summarizes the demographics and characteristics of the disease. The mean age of the recruited patients was 43 years with a higher proportion of males (86%), with a mean Body Mass Index of 25, and a median ASA classification of II (63%). The principal cause of ON was idiopathic (50%), followed by corticosteroid treatment (22%). Sixty-eight percent of the affected femoral heads were classified as ARCO IIA, and the main evolution of ON prior to treatment was 2.3 ± 2.2 months. No statistically significant differences between cases that did not heal (progression and/or THR) and those that healed were observed in the baseline, in terms of age (*t*-test = 0.471), BMI (*t*-test *p* = 0.658), sex (Fisher’s test *p* = 0.905), ASA classification (Fisher’s exact tests *p* = 0.247), alcohol consumption history (Fisher’s exact tests *p* = 0.477), smoking habit history (Fisher’s exact tests *p* = 0.500), etiology of ON (Fisher’s exact test *p* = 0.623), time of ON evolution (Mann–Whitney test *p* = 0.749), ARCO classification (Fisher’s exact test *p* = 0.132), preoperative loss of bone (Fisher’s exact test *p* = 0.470), Total Harris Hip Score (HHS) (Mann–Whitney test *p* = 0.783), spontaneous pain (Mann–Whitney test *p* = 0.106), and weight-bearing pain (Mann–Whitney test *p* = 0.152).

Sphericity was observed at inclusion in all recruited cases. No history of drug consumption was declared. At the recruitment visit, one patient had a history of immunosuppressive treatment, and eight patients had had occasional corticosteroid treatment, neither interpreted as the cause of ON.

### 2.2. Study Authorization and Declaration

Four Ethics Committees (EC) of clinical research in the four participating countries (Person Protection Committee (Comité de Protection des Personnes CPP) Tours Région Centre Ouest 1, Tours, France; La Paz Hospital Ethics Committee for Clinical Research (Comité de Ética de la Investigación clínica CEIC), Madrid, Spain; Ulm University Ethics Committee, Ulm, Germany; and Istituto Ortopedico Rizzoli Ethics Committee, Bologna, Italy) approved the protocol and related documents for all participating clinical centres. The Ethics Committee at La Paz Hospital, the coordinator centre, gave the final approval on 24 June 2013, with hospital code 3875. As the investigational medicinal product (IMP) was an Advanced Therapy Medicinal Product (ATMP) for human use, the responsibility to authorize the trial relies on the National Competent Authorities (NCA), following the European Voluntary Harmonization Procedure (VHP, VHP201332). The authorizations were obtained in all participating countries between January and March 2013. The sponsor of the study was the Autonomous University of Madrid, Madrid, Spain. The EudraCT number of the trial was 2012-002010-39 and the trial was also incorporated into the database ClinicalTrials.gov with the identifier NCT02065167.

### 2.3. Procedures

#### 2.3.1. Bone Marrow Harvesting

The donation, procurement and testing of the BM were performed in compliance with the Cells and Tissues Directives; in particular, according to the requirements laid down in Directive 2006/17/EC of 8 February 2006 implementing Directive 2004/23/EC of the European Parliament and of the Council as regards certain technical requirements for the donation, procurement and testing of human tissues and cells, and applicable national laws. Specifically, patients needed to be negative in serology for Anti-HIV 1-2 Ab, Anti-HCV Ab, HBs Ag, Anti-HBc Syphilis, and negative (not detected by PCR) in HIV NAT, HCV NAT, or HBV NAT. The bone marrow cells were harvested in the operating room under anaesthesia, from the posterior iliac crest, through percutaneous bone puncture. Bone marrow was harvested by fractions of 2–4 mL in 20 mL syringes prefilled with heparin, then transferred into a transportation bag, and labelled according to the approved protocol. The harvest, in its primary packaging, was laid out in an isothermal box labelled according to Directive 2004/23/EC and 2006/17/EC. The transport was done between 18 °C and 24 °C if less than 30 min, and at 4 °C with temperature traceability if the transportation time was longer than 30 min.

#### 2.3.2. Cell Product Manufacturing Process in GMP Facilities

An aliquot of the starting material was removed to carry out controls, including cell count, viability, CFU-F-assay, and sterility. The entire manufacturing process was performed at each manufacturing site, following the same procedure described in ORTHO-1 CT (EudraCT 2011-005441-13) and ORTHOUNION CT (EudraCT 2015-000431-32) [14,15]. In brief, the culture was performed when the received BM, without any further manipulation, was seeded in alpha-MEM medium with 5% PL and 1 I.U./mL heparin, at the concentration of 50.000 WBC/cm^2^, in a culture chamber. The culture chambers were placed in incubators (5% CO_2_ atmosphere, 95% relative humidity at 37 °C). After 72 h, the supernatant was discarded and replaced by fresh complete medium (alpha-MEM with 5% PL). At day +7 and +10 of culture, the supernatant was again discarded and replaced by complete medium. At +14 day, the confluence was evaluated and if >50%, the cells were washed with phosphate buffered saline (PBS), detached and harvested using trypsin. If cell confluence was lower than 50%, an additional medium exchange was performed, and cells were harvested at day 17. The harvested cells were then re-seeded at the concentration of 4 × 10^3^ MSC per cm^2^ in new culture chambers in alpha-MEM medium with 8% PL. Another medium exchange was performed at day 17. At day 21, the cell culture was washed with PBS and the cells detached and harvested using trypsin. The final product resulting in ORTHO-2 BM-MSC was the active substance on which quality controls were applied. The active substance was resuspended in an albumin solution to obtain the ORTHO-2 BM-MSC tissue-engineered product. Cells were packaged for the shipment to the operating room in 1 syringe of 7 mL, at a dose of 20 million cells per mL (total of 140 million cells). All the materials and reagents were selected and validated to ascertain their compliance to be used in the manufacturing process, with certificates of analysis of key components included in the investigational medicinal product (IMP) for which approval was obtained at each of the NCA of the participating countries (France, Germany, Italy, Spain).

The cell expansion process and the delivered final cell product is described in Table 2, including data from the bone marrow aspiration, at seeding, reseeding at P0, reseeding at P1, and product release criteria. The achieved standardization of the cell product can be observed in Table 2.

**Microbial safety:** 100% of endotoxin tests and 100% of mycoplasma tests at P1 were negative (n = 10). Microbial tests were negative (n = 25) in BM samples (100%), in expanded cells at P0 (96%) and expanded cells at P1 (100%). The production process of the contaminated sample identified at P0 was stopped and material destroyed.

#### 2.3.3. Cell Product

The IMP was composed of mesenchymal stromal cells (MSC) obtained through a manufacturing process based on plastic adherence and expanded in culture using 5% human donor platelet lysate produced in Ulm (Germany) and distributed to the other cell therapy units, according to a culture method developed by the REBORNE consortium. MSCs were defined by a specific immunophenotype (CD 45−/90+/105+/73+/HLA-DR) [16] with demonstrated osteogenic properties in vitro and in vivo. The IMP manufacturing authorization was granted to all five participating GMP facilities (Établissement Français du Sang (EFS) at Créteil and Toulouse, in France; Transfusion Medicine Institute of Ulm in Germany; Fondazione IRCCS Ca’ Granda Ospedale Maggiore Policlinico of Milano in Italy; and Cell Production Unit at Hospital Puerta de Hierro-Majadahonda of Madrid in Spain). Each batch of the final product was tested for cell content, immunophenotype, sterility, endotoxins, and Mycoplasm before release. Additional quality controls were performed according to each country-specific national competent requests. Descriptive cell values, along the process of expansion and at the release of the cell product in the treated cases, are included in Table 2. A suitable mode of transportation ensured the delivery of the BM package to the manufacturing site and of the cells from the GMP facility to the surgical room within 18 h, and the process was validated for cell viability [10,13].

#### 2.3.4. Surgical Procedure

As with any other implant in Orthopaedic surgery and to avoid any risk of bacteraemia, antibiotic prophylaxis according to the protocol of each hospital (such as cephazolin 1 g iv preoperative) was performed prior to the procedure. After anaesthesia, patients were positioned supine on a fracture table under sufficient traction to maintain the patient lower limbs. A radiological C-arm was placed and both AP and axial views of the femoral head and neck were checked under fluoroscopy.

The surgical approach was minimally invasive lateral to the proximal femur. A guide wire was drilled from the lateral cortex of the subtrochanteric femur into the femoral head lesion, under fluoroscopic AP and axial control. Then, a 4 mm cannulated drill was introduced along the drilling guide into the femoral head (there was no notification of femoral head cartilage perforation by drilling). The cells (140 million cells suspended in 7 mL) were injected directly in the forage tunnel in a single administration. Drains were not used in this procedure.

### 2.4. Outcomes

#### 2.4.1. Safety

A primary safety endpoint, defined as detection of local and general complications, was fixed for the clinical trial at any time in the 24 months of follow-up; adverse event (AE) reporting at 3, 6 and 12 months; severe adverse event (SAE) and suspected unexpected and serious adverse reactions (SUSAR) reporting at any time, as required by the regulatory frame (to Eudravigilance (European Union Pharmacovigilance database), to the National Competent Authorities and the Ethics Committees).

#### 2.4.2. Efficacy

The efficacy was defined as bone healing, without increasing the complication rate, of early osteonecrosis treated through a standard-of-care core decompression procedure plus a percutaneous injection of autologous stem cells, derived from bone marrow and expanded under GMP conditions. Non-healing was defined as radiological (X-ray or MRI) progression to a higher stage and/or undergoing Total Hip Replacement (THR).

Radiological progression on X-ray (anteroposterior and lateral views) was evaluated at 3, 6 and 12 months, and on MRI (coronal and transversal view) at 3 and 6 months. The long-term radiological progression was assessed on X-rays at 5 to 6 years. Clinical healing was considered when the pain was under the threshold of 30 (out of 100 in a Visual Analog Scale (VAS)) [17], and the imaging studies did not show any progression or collapse of the femoral head.

### 2.5. Statistical Analysis

Descriptive statistics and Fisher’s exact test or Mann–Whitney test were used to compare categorical and continue baseline variables. Comparison of means was performed with paired Student’s *t*-test between follow-up visits for pain and Harris Hip Score variables. A generalized estimating equation (GEE) with logit link function was conducted to evaluate the risk factors of ON-progression or THR. The statistical significance was defined with 95% of confidence (*p* ≤ 0.05). Data were analysed using STATA software version 12 (StataCorp., College Station, TX, USA).

## 3. Results

### 3.1. Safety Endpoint

No Adverse Event (AE), or Serious Adverse Event (SAE), or Suspected Serious Adverse Reactions (SUSAR), were identified as related with the IMP. Particularly, no tumorous condition or cell-related overgrowth was detected in any patient after cell implantation. A total of 4 SAEs were communicated, unrelated to IMP, and requiring inpatient admission and treatment. The first SAE was in a 43-year-old female, with one year of evolution intermediate thalassemia, who presented with deep venous thrombosis one month after surgery, for a reported duration of 25 days. The second SAE was in a 48-year-old male who presented with contralateral femur fracture 3 months after surgery while playing volleyball. He was treated by open reduction and internal fixation, discharged two days after fracture surgery, and fully recovered without sequelae. The third SAE occurred in a 34-year-old male, admitted 8 months after the index operation due to acute arthritis, received antibiotics despite negative cultures, and resolved after THR due to persistent pain, thus being considered a non-healed case. The fourth SAE occurred in a 45-year-old male who required hospital admittance due to upper gastrointestinal bleeding, 12 months after surgery, that fully resolved at 3 days. Reported AEs included persistent pain, pain and limping, and pain at the contralateral leg.

### 3.2. Efficacy Endpoint and Follow-Up

Mean preoperative spontaneous pain was 30.3 ± 20.5 mm on theVAS scale, and changed to 15.4 ± 18.1 mm at three months of FU (paired *t*-test *p* = 0.018), to 15.3 ± 26.3 mm at 6 months of FU (paired *t*-test *p* = 0.013), and to 9.5 ± 17.8 mm at 12 months of FU (paired *t*-test *p* = 0.001). Mean preoperative weight-bearing pain was 57.9 ± 21.1 mm on the VAS scale, and changed to 30.5 ± 24.2 mm at three months of FU (paired *t*-test *p* = 0.018), to 32.8 ± 31.6 mm at 6 months of FU (paired *t*-test *p* = 0.013), and to 26.0 ± 21.3 mm at 12 months of FU (paired *t*-test *p* = 0.014). Total Harris Hip Score improved a total of 19.6 ± 16.8 points from preoperative to 12 months FU, which was statistically significant (paired *t*-test *p* = 0.008).

Clinical and radiological regeneration (Figure 2), with maintained head sphericity, was observed in 80% of the treated patients (16/20) after a one-year follow-up, 73% (16/22) on ITT (including 2 drop-out cases only evaluated at 3 months). Four cases progressed to the next stage of ON (4/20), and of these, 3 received THR. Generalized estimating equation analysis (model adjustment: Wald Chi-square = 177.35; *p* = 0.001) for pain higher than 30/100, age higher than 50, time since diagnosis higher than 3 months, positive alcohol consumption, and positive smoking habit, only showed a significant interaction of ON progression/THR with pain (OR = 3.8 [3.0–4.7]; *p* = 0.001).

After 5 to 6 years of follow-up, neither other case had progressed to the next stage of ON nor received another THR. We investigated 22 cases, including drop-out cases. One patient was not located at 5 years and was lost after completing the 2-year FU in the trial but was considered healed at 2 years. Of the remaining 21 cases, we observed that 16/21 were healed and maintained their femoral head at 5 years, 1 had progressed to the next ON stage in the first year of treatment but required no THR at 5 years, 4 cases in total had received THR (including one of the drop-out cases), all in the first year after surgery. Details of the case progression are described in Figure 3.

## 4. Discussion

The main result of this study is that bone regeneration in the femoral head can be safely obtained with autologous expanded MSCs from bone marrow at a dose of 140 million MSCs, injected into the femoral head through a minimally invasive approach after forage. This confirms findings in many other reports and reviews about cell therapy to regenerate bone with expanded [18] or concentrated MSCs [9,19,20].

Regarding the efficacy outcome, the literature on MSCs to treat early ON offers ranges of 70–90% avoidance of collapse and/or THR at 2 to 5 years, and even 30 years [8,9,19,20,21]. Different sources of variability have been identified, related to patients, diagnosis, and treatment. All our cases were acute (under 8 months of evolution since the onset of the disease), and treatment was not performed on established long-term osteonecrosis. Late treatment after the ON onset may have encountered spontaneous repair in the periphery of the necrosis, preventing the collapse [1], which may be unrelated to the treatment under study. Although seldom reported, the onset of the ON lesion under treatment should be clarified to compare the outcome of different ON treatments and series. Another source of variable outcome is the ON aetiology, as corticosteroid-related ON (5/22 in our trial) may associate less locally available osteoprogenitors [22], and the cellular reservoir is unclear in other forms of ON, particularly if idiopathic. The indication of THR also incorporates variability into the outcome, as patients may request early THR even if no head collapse has occurred but if persistent early pain is not controlled. Surgical indication of THR may vary centre to centre, and restrictive THR indication was identified as a cause of one early drop-out that received THR in a different centre after leaving the study, without documented head collapse. Of note, all THRs were performed in the first year after MSC treatment, thus confirming that bone regeneration, when obtained, was stable and successfully healed patients that would not require any other operation at least in 5 years. Instead, other studies showed an increasing number of THRs along the 5 years follow-up [23], which may relate to less efficacious regeneration, prompting some researchers to advocate biomechanical augmentation instead of cell therapy alone [24].

Bone regeneration was observed in all our cases after receiving the expanded cells, even if THR was required in 19% of the cases at 5 years (3/22 received THR in the trial, plus one more THR in an early drop-out). Other studies have shown that 70% of the asymptomatic hips with no treatment became either symptomatic (38%) or collapsed (32%) after 5 years or more [25]. Besides, forage alone may be effective to decrease pain immediately after surgery [26], but not to avoid THR in almost half of the cases in stage II ON [6,27]. Results with bone marrow concentration injected through the forage in early ON stages favorably compared with forage alone [28,29,30], although this may change depending on the technique and the patient [27]. Taken together, the success rate of Bone Marrow Concentrate (BMC) injection after forage, compared with forage alone, is confirmed in clinical scoring, head collapse and THR conversion, through a meta-analysis of the available trials [20], although serious variability and risk of bias were also detected.

The question remains whether expanded MSC can perform better and more predictably than BMC. However, data on expanded MSC are scarce [11,31]. A dose of 2 × 10^6^ expanded cells was considered to control the progression of 51/53 injected hips at 5 years versus 27/44 (with 7 lost to FU) hips after forage [11], which can be considered encouraging results. Unfortunately, the expansion procedure, quality controls and trial procedure are unclear in early reports [11]. As an augmentation of tricortical iliac bone graft with vascular pedicle and biomaterial, expanded MSC were implanted in another study at a dose of 0.5–1 × 10^8^ [31]. Cells were not expanded under GMP criteria and were frozen and thawed before implantation. While no progression was observed at 2 years in 7 out of 9 hips with ON stage III, the real effect of cells is unclear [31]. In our study with 140 × 10^6^ expanded MSCs, 3 THR and 1 progression in the ON stage were identified out of 20 cases, but no progression was seen after 12 months until the latest follow-up of minimum 5 years.

Pain and clinical scoring were significantly improved by the technique, and therefore can be seen as a significant predictor of failure (progression and/or THR), although our limited number of patients may require further confirmation. Unfortunately, pain was not controlled and/or osteonecrosis progression was not necessarily stopped by the obtained bone regeneration in those failed cases. Neither time since diagnosis nor alcohol consumption, smoking habit, or age showed any association with ON progression in our study. Therefore, other aspects may influence the early progression to head collapse, possibly including the amount, timing, and location of the obtained bone regeneration. It is unclear if those aspects may relate to the disease, the patient, or the surgery characteristics. Further studies are required to clarify if the disease aetiology, the patient regenerative potential or the surgical technique may be at the origin of this progression.

Remarkable aspects in the study include the clinical use of a high dose of expanded autologous MSC (140-million cells total dose), higher than other reported trials. Interestingly, a cell dose of 1 × 10^8^ (100 million cells) has been defined in a meta-analysis as a threshold for better effects on disease progression [32]. This dose safely obtained the clinical and radiological healing of femoral head in our study, both in males and females, despite the limited number of cases in such an early trial. The relationship between dose and healing success has been previously proven with bone marrow concentrate [33], and this is probably also the case with expanded cells.

The first limitation of the study is the design, while a comparative, randomized study would offer higher evidence. However, strength of the design is the multicentric, multinational set-up of the clinical trial that validates the approach and the development of a consistent expanded-cell medication in different settings. Also, the long follow-up (at least 5 years) is a strength of the current study, as it is rare in a complex clinical trial on this kind of treatment. Another potential limitation is the variability related to the origin and cause of ON, and to the patient, as the trial evaluates an autologous treatment with a different osteogenic potential of individual patients. However, the stringent inclusion/exclusion criteria support the study validity and relevance. Also, a limitation relates to the difficult endpoint of efficacy, here defined as ON progression, eventually to collapse, and/or THR. Timely bone regeneration in a sufficient amount to avoid progression depends not only on the cells but also on the status of the disease and the implanted femoral head, while THR indication also depends on the patient tolerance to pain. Therefore, other means of calculating efficacy may be required.

## 5. Conclusions

We showed in a multicentric interventional trial that the administration of high-dose MSCs, expanded from autologous bone marrow, was safe and capable of producing bone regeneration in all studied cases at 3 and 6 months. Also, maintenance of femoral head sphericity was confirmed per protocol in 16/20 (80%) of treated hips at one year. Finally, a minimum 5 years follow-up, including early drop-outs, could confirm that no THR or ON stage progression was detected after 12 months, and head sphericity was maintained in 16/21 cases at 5 years.

Although a definite proof of efficacy would require a randomized trial, we controlled and highlighted different crucial issues, such as disease variability (only acute and subacute cases were recruited), osteogenic potential (unclear differences due to autologous origin), THR indication (restricted indication if no collapse), and we provided long-term follow-up to clarify whether the ON was successfully healed.

## Figures and Tables

**Figure 1 jcm-10-00508-f001:**
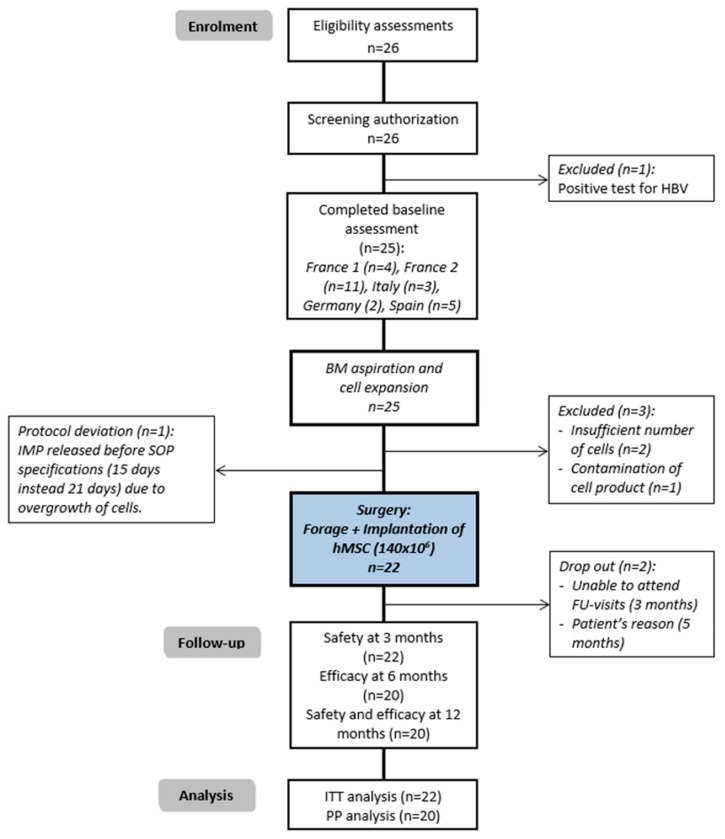
CONSORT diagram of the ORTHO2 clinical trial. ON: Osteonecrosis; BM: Bone Marrow; IMP: Investigational Medical Product; hMSC: Human Mesenchymal Stems Cells (expanded).

**Figure 2 jcm-10-00508-f002:**
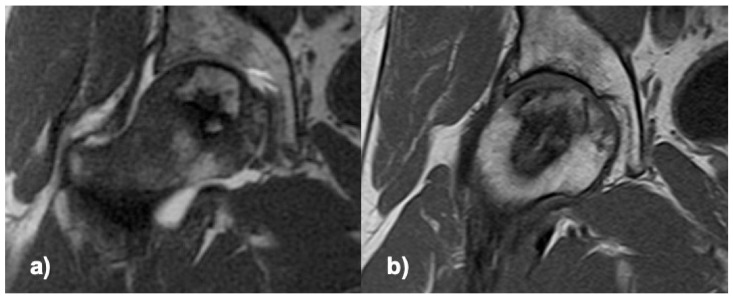
T1 MRI of the femoral head, coronal section through the ON area; (**a**) preoperative, (**b**) 6 months after delivering the cells.

**Figure 3 jcm-10-00508-f003:**
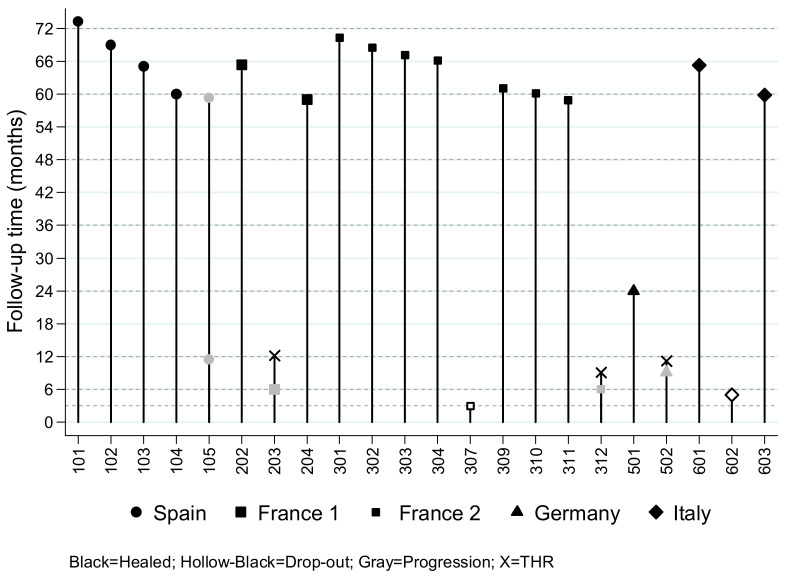
Late follow-up of ORTHO2 patients (last update, January 2020): One case with early progression to the next stage without collapse after five years (case 105). Three cases of THR during the first year of the clinical trial (cases 203, 312, 502). One case drop-out at 3 months (case 307), healing confirmed at 5 years. One case drop-out at 5 months (case 602), underwent THR in the first year. One case healed at 2 years (case 501), lost to follow-up at 5 years.

**Table 1 jcm-10-00508-t001:** Demographics and characteristics of the disease (n = 22).

Variables	Mean n	SD %	Min	Max
**Demographics**				
Age (years)	43.1	10.9	21.0	62.0
Height (cm)	173.7	8.7	152	190
Weight (Kg)	77.4	20.0	50	118
BMI	25.7	5.6	17.3	36.8
Male sex	19	86%		
**ASA:**				
*I*	5	22.7%		
*II*	14	63.6%		
*III*	3	12.6%		
**Alcohol history (yes)**	7	32%		
No. drinks per day ^1^	2.2	1.9	1	6
Duration (years) ^2^	12.8	6.9	2	20
**History of smoking (yes)**	11	50%		
No. packs per day	1.1	0.7	0.2	2
Duration (years) ^3^	19.8	8.8	5	30
**Months since diagnosis of ONFH ^4^**	2.3	2.2	0.1	7.6
**Cause of ONFH:**				
*Alcohol consumption*	1	4.5%		
*Corticosteroids*	5	22.7%		
*Idiopathic*	11	50.0%		
*Other ^5^*	5	22.7%		
**Description of the affected femur**				
Laterality (Right)	13	59%		
ARCO classification:				
*IIA*	15	68.2%		
*IIB*	6	27.3%		
*IIC*	1	4.5%		
Loss of bone in AP X-ray (none) ^6^	18	82%		
**Preoperative clinical data**				
Total Harris Hip Score	63.0	19.0	40	91
Spontaneous pain in VAS scale (mm) ^7^	30.3	20.5	1	76
Weight-bearing pain in VAS scale (mm) ^8^	57.9	21.1	10	100

ONFH: Osteonecrosis of the Femoral Head; VAS: Visual Analogue Scale; ^1^ One drink corresponds to 150 mL of wine or 360 mL of beer or 45 mL of hard liquor.; ^2^ One case stopped drinking 11 years ago; ^3^ Three cases reported being former smokers since 1, 6 and 7 years.; ^4^ Median (P25/P75) = 1.5 (0.5/3.4) months; ^5^ Alcohol+Corticosteroid (n = 1) Sickle cell disease (n = 2), Thalassemia (n = 1), Octreotide treatment (n = 1); ^6^ Four cases reported a loss of bone minor than 50% of diameter; ^7^ Median (P25/P75) = 29.5 (17.0/41.2) months; ^8^ Median (P25/P75) = 60.0 (48.0/70.0) months.

**Table 2 jcm-10-00508-t002:** Cell expansion and final cell product.

	Mean	SD	Min	Max
**BM aspiration**				
Aspirated bone marrow volume	51.20	8.80	31.50	65.00
Cell count WBC/mL BM aspirate (×10^7^)	2.07	11.40	11.20	61.00
CFU-F of BM/×10^6^ WBC	45	53	0	190
**Doubling time and population doubling in p0 and p1 of expansion**				
Doubling time in P0 (h)	23.80	3.30	12.40	28.20
Doubling time in P1 (h)	54.80	31.50	22.10	186.00
Number if population doublings in P0	14.60	3.00	12.40	27.10
Number if population doublings in P1	3.20	1.20	0.90	6.50
Cumulative population	17.70	3.40	13.80	30.00
**Yield**				
MSC/ul BM aspirate in P0 (×10^3^)	8.46	10.80	0.24	51.00
MSC/ul BM aspirate in P1 (×10^4^)z	6.90	9.31	0.21	35.20
Overall harvest (×10^8^)	2.87	1.87	0.42	7.38
**Identity: Surface markets after P1**				
%CD34 positive cells	0.29	0.26	−0.09	1.30
%CD45 positive cells	0.49	1.14	0.00	5.40
%CD73 positive cells	97.81	4.77	78.00	100.00
%CD90 positive cells	99.30	0.49	93.20	100.00
%CD105 positive cells	97.38	4.20	82.40	100.00
%MHC cII positive cells	3.62	7.01	0.00	27.60
**Viability**				
% Viable cells in aspirate	95.37	3.20	89.40	99.99
% Viable cells after P0	94.92	4.70	82.70	100.00
% Viable cells after P1	96.29	2.82	88.30	99.90

## Data Availability

Anonymized data available on request due to ethical restrictions.

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
