# Peer review of "Osteonecrosis of the Femoral Head Safely Healed with Autologous, Expanded, Bone Marrow-Derived Mesenchymal Stromal Cells in a Multicentric Trial with Minimum 5 Years Follow-Up"

_jcm, 2021, doi:10.3390/jcm10030508_

Round 1

Reviewer 1 Report

As mentioned by authors in his discussion, this paper could has more clinical
evidence if had a control group.

But is a good paper showing alternative solutions to the classic osteonecrosis therapies. Study design is good and relevant.

Author Response

Thank you for your realistic view on our paper. Our answers:

As mentioned by authors in his discussion, this paper could has more clinical
evidence if had a control group.

Thank you for your understanding. In fact, this is a Phase I/IIa clinical trial and this is why it was not designed with a control group. Besides, a good control group should be possibly very restrictive (etiology, classification, first 6 months after onset,...), but we agree this is a challenge for the future.

But is a good paper showing alternative solutions to the classic osteonecrosis therapies. Study design is good and relevant.

Thank you for your consideration. Indeed we feel cell therapy solutions need significant efforts to obtain standardisation and to perform adequate studies, and we tried to do so.

Also, following your suggestion, we performed an editing of English language and style, with revision from colleagues with native and fluent English. We hope to have improved the manuscript also in this sense.

Reviewer 2 Report

Thank you for the opportunity of reviewing this work. i think it is well written and designed. The primary endopoint has been successfully reached. I trust this technique in the treatment of AVNFH that remains still challenging. Nevertheless I have some questions for you:

  • can you specify the duration of non weight bearing after surgical procedure? (if actually prescribed)
  • can you explain the absence of ARCO stage I  patients in your study?
  • can you clarify if the progression towards THA or loss of sphericity has been shown by patients in stage IIb and IIC? Of course, in early stages of ON also CD alone could be more effective but this finding should be consistent with literature you cited ( Aigner, N.; Schneider, W.; Eberl, V.; Knahr, K. Core decompression in early stages of femoral head osteonecrosis--an MRI-503 controlled study. International orthopaedics 2002, 26, 31-35.).

Author Response

Thank you for your efforts reviewing our paper. Our answers follow:

Thank you for the opportunity of reviewing this work. i think it is well written and designed. The primary endopoint has been successfully reached. I trust this technique in the treatment of AVNFH that remains still challenging.

Thank you for your kind words. We feel an interesting field is opened with expanded BM-MSCs due to standardisation and high number of cells, and we are confident we have learned important lessons from this study.

Nevertheless I have some questions for you:

  • can you specify the duration of non weight bearing after surgical procedure? (if actually prescribed).

Non weight bearing was recommended for the first 3 weeks after surgery, followed by partial weight bearing 3 more weeks. However, this was not included in the protocol and some variability could occur among centres. There is no consensus about postoperative weight bearing after core decompression in hip osteonecrosis, and this is why it was not introduced, but we feel this fact did not influence the progression of osteonecrosis in our cases as most of our patients have avoided unilateral weight bearing postoperatively.

  • can you explain the absence of ARCO stage I  patients in your study?

The study recruitment was performed with symptomatic patients with MR required. This is probably why stage I, less symptomatic or without sufficient suspiction to have an early MR, was less frequent. But the advantage is that this selection allows us to conclude about ARCO II patients.

  • can you clarify if the progression towards THA or loss of sphericity has been shown by patients in stage IIb and IIC? Of course, in early stages of ON also CD alone could be more effective but this finding should be consistent with literature you cited ( Aigner, N.; Schneider, W.; Eberl, V.; Knahr, K. Core decompression in early stages of femoral head osteonecrosis--an MRI-503 controlled study. International orthopaedics 2002, 26, 31-35.).

The progression of osteonecrosis (stage progression or THA) has been found in IIB and IIC in our study. However, we investigated the difference in the ARCO classification between healed and non-healed cases and we could not find a statistical significance (Fisher’s exact test p=0.132). To provide more information about this issue, a paragraph has been added with the comparison of demographics and disease characteristics between healed and non-healed cases, including the ARCO classification (IIA, IIB, IIC). The concentration of our cases in ARCO II may produce this lack of significance, whereas ARCO III cases included in some CD studies confirm than higher ARCO stages are associated to treatment failure (in line with Aigner et al. 2002, treating with CD stages I to III). However, of note, a recent consensus (Delphi approach) on ARCO classification (Yoon BH, Mont MA, Koo KH, Chen CH, Cheng EY, Cui Q, et al. The 2019 Revised Version of Association Research Circulation Osseous Staging System of Osteonecrosis of the Femoral Head. J Arthropl.2020;35(4):933-40) is favorable to separate ARCO III categories, but not ARCO II. We agree this can be a matter of debate and more cases would be required to clarify this issue.

Reviewer 3 Report

This work shows the results of  European multicentric clinical trial which demonstrate safety and early efficacy of in vitro expanded bone marrow-derived MSCs for the treatment of early femoral head Osteonecrosis. The accuracy of the study is high as well as the selection of patients. Methods are well described and according to GMP regulamentation. An extremely relevant aspect of this work is the long term follow-up evaluation. Of course one limit is the small number of patients, but this is a phase I/IIa study, thus it is acceptable. The other limitations of the study have been correctly discussed in the manuscript. I believe that these results are extremely useful for future clinical studies and they are suitable for publication.

Author Response

We thank the reviewer for his/her review of our manuscript.

This work shows the results of European multicentric clinical trial which demonstrate safety and early efficacy of in vitro expanded bone marrow-derived MSCs for the treatment of early femoral head Osteonecrosis.

Thank you for your appreciation. Indeed, it has been a conjoint effort through a complex multicentric, multinational clinical trial, and this is why the list of involved researchers is considerable, as listed in the Acknowledgments. We would like also to thank all of these researchers, with substantial contributions to the trial.

The accuracy of the study is high as well as the selection of patients. Methods are well described and according to GMP regulamentation. An extremely relevant aspect of this work is the long term follow-up evaluation.

Thank you for these words. The clinical trial was planned for 2 years, but we differ the publication until 5 years follow-up was completed because we fully agree in that the long-term follow-up is key to consider any cell therapy and any femoral head preserving treatment.

Of course one limit is the small number of patients, but this is a phase I/IIa study, thus it is acceptable. The other limitations of the study have been correctly discussed in the manuscript.

We are fully aware of the limitations of such a trial and we appreciate the positive comment from the reviewer.

I believe that these results are extremely useful for future clinical studies and they are suitable for publication.

Thank you very much for these supportive words, we feel other future clinical studies on the topic may find useful information in this work.